# Associations of Bone Mineral Density with RANKL and Osteoprotegerin in Arab Postmenopausal Women: A Cross-Sectional Study

**DOI:** 10.3390/medicina58080976

**Published:** 2022-07-22

**Authors:** Osama E. Amer, Kaiser Wani, Mohammed G. A. Ansari, Abdullah M. Alnaami, Naji Aljohani, Saba Abdi, Syed D. Hussain, Nasser M. Al-Daghri, Majed S. Alokail

**Affiliations:** 1Chair for Biomarkers of Chronic Diseases, Biochemistry Department, College of Science, King Saud University, Riyadh 11451, Saudi Arabia; osamaemam@gmail.com (O.E.A.); wani.kaiser@gmail.com (K.W.); ansari.bio1@gmail.com (M.G.A.A.); aalnaami@yahoo.com (A.M.A.); sabdi@ksu.edu.sa (S.A.); danishhussain121@gmail.com (S.D.H.); 2Obesity, Endocrine and Metabolic Center, King Fahad Medical City, Riyadh 59046, Saudi Arabia; najij@hotmail.com; 3Protein Research Chair, Biochemistry Department, College of Science, King Saud University, Riyadh 11451, Saudi Arabia; malokail@ksu.edu.sa

**Keywords:** BMD, PMO, bone markers, osteoprotegerin, osteoporosis, RANKL

## Abstract

*Background and objective*: There is limited information as to the association of several key bone markers with bone mineral density (BMD) in understudied ethnic groups. This study investigated the relationship between circulating levels of osteoprotegerin (OPG) and receptor activator of nuclear factor kappa-Β ligand (RANKL) with BMD in Arab postmenopausal women. *Materials and methods*: In this cross-sectional study, a total of 617 Saudi postmenopausal women from the Osteoporosis Registry of the Chair for Biomarkers of Chronic Diseases were included. Anthropometric data, BMD, and biochemical data were retrieved from the registry. Participants were stratified into three groups based on T-score; *n* = 169 with osteoporosis, *n* = 282 with osteopenia, and *n* = 166 normal. Analysis of bone markers including RANKL, OPG, osteocalcin, and N-terminal telopeptide (NTx) was completed using commercially available bioassays. *Results*: The results suggested that OPG was significantly and positively correlated with age in the osteoporosis group (*r* = 0.29, *p* < 0.05), while it was inversely correlated with BMD femoral neck left (r = −0.56, *p* < 0.001) and BMD femoral neck right (*r* = −0.37, *p* < 0.05) in the same group. Moreover, RANKL showed a significant inverse correlation with NTx in the osteopenia group (*r* = −0.37, *p* < 0.05). Furthermore, the RANKL/OPG ratio had a positive and significant correlation with BMI (*r* = 0.34, *p* < 0.05), BMD femoral neck left (*r* = 0.36, *p* < 0.05) and BMD femoral neck right (*r* = 0.35, *p* < 0.05) in the osteopenia group. By contrast, it showed a significant inverse correlation with waist to hip ratio in the osteoporosis group (*r* = −0.38, *p* < 0.05). Multiple regression analysis showed that OPG contributes to BMD variations in the osteopenia group (*p* = 0.03). *Conclusions*: In conclusion, changes in circulating levels of RANKL and OPG might be a protective mechanism contrary to the increased bone loss in postmenopausal women.

## 1. Introduction

Osteoporosis is a condition characterized by weakened bone strength which increases the likelihood of fractures [1]. Increased bone loss, cumulative bone turnover, and subsequent bone fractures are serious consequences of postmenopausal osteoporosis in women, resulting in a lower overall quality of life and a higher mortality rate [2]. The identification of a receptor activator of nuclear factor-kappa B ligand (RANKL), a critical cytokine for osteoclastogenesis, has advanced our understanding of bone physiology [3,4,5]. RANKL, expressed on osteoblasts, acts as a transmembrane ligand, causing osteoclast differentiation, activation, and inhibition of osteoclast apoptosis, resulting in increased bone resorption and bone loss [5,6]. This RANKL bone resorption effect can be mitigated by osteoprotegerin (OPG), a RANKL decoy receptor that is thought to be a key inhibitory player in bone resorption [3,7]. 

Studies revealed that the RANKL to OPG ratio contributes to bone remodeling [8,9,10]. When OPG is higher than RANKL, bone formation occurs, while the opposite of this OPG and RANKL relationship favors bone resorption [11,12]. As a result, OPG can protect bones by neutralizing RANKL’s osteoclastic properties [12,13]. In addition to osteoblasts, RANKL and OPG are also produced in other tissues such as adipose allowing bone metabolism [14]. As a result, measuring the RANKL/OPG ratio may reflect levels from both non-skeletal and skeletal sources [15]. 

The role of the immune system in the development of osteoporosis in postmenopausal women, primarily caused by estrogen deficiency and secondary hyperparathyroidism, is being studied [16,17]. Besides, other hormones, such as testosterone and estradiol, may influence the balance of RANKL and OPG [18,19]. Nevertheless, the pathophysiological implications in the variations of circulating RANKL and OPG are not entirely understood [20]. Several human studies have found inconsistencies in the relationship between serum OPG and bone mineral density (BMD) as well as bone biomarkers [21,22,23]. While serum RANKL changes have been reported in multiple myeloma [23], Paget’s disease [24], and rheumatoid arthritis (RA) [25], limited studies are reported in postmenopausal women where the correlations between RANKL and OPG with age and BMD have been investigated, particularly in Arabian ethnic groups such as Saudi Arabia where the prevalence of osteoporosis is increasing [26]. The purpose of this study was thus to investigate the circulating levels of OPG and RANKL and their associations with BMD and look into other factors that influence these markers in Arab postmenopausal women.

## 2. Materials and Methods

### 2.1. Subjects

In this cross-sectional study, 617 postmenopausal Saudi women were included. Clinical information was obtained from the Osteoporosis Registry of the Chair for Biomarkers of Chronic Diseases (CBCD) in King Saud University, Riyadh, Saudi Arabia [27]. In brief, the osteoporosis registry is a database of consenting Saudi men and women aged 35 and above, recruited from different primary health care centers in Riyadh, Saudi Arabia. Bone mineral density (BMD) (g/cm^2^) at the lumbar spine, left hip, and right hip was assessed using DEXA (Hologic QDR 2000 Inc., Woltham, MA, USA). Participants were stratified into three groups: 169 with osteoporosis, 282 with osteopenia, and 166 with normal postmenopausal women. Diagnosis of osteoporosis as ones with T-score of or less than −2.5 standard deviations (SD) was based on the definition used by the regional guidelines [28]. Similarly, osteopenia was defined as T-score between −2.5 and −1.0 while the participants with T-score above −1.0 were considered having normal BMD. Anthropometry, height, weight, waist and hip circumferences, and systolic and diastolic blood pressure were noted. Body mass index (BMI) was calculated as weight (kg)/height (m^2^). The inclusion criterion for the current study was Saudi women in CBCD osteoporosis registry who have reached the state of menopause. The exclusion criteria included women with pre-existing bone diseases other than low BMD or under medications that can affect BMD and/or bone turnover markers for at least 6 months. Specifically, women taking medications such as non-steroidal anti-inflammatory drugs, prostaglandin E and zoledronic acid, as well as those who had a history of treatment with pulsed electromagnetic fields (PEMFs), known to modulate RANKL/OPG system, were excluded [29]. Furthermore, those under vitamin D or calcium supplementation were also excluded. This study was conducted according to the guidelines of the Declaration of Helsinki and approved by the Ethics Committee of the College of Science, King Saud University, Riyadh, Kingdom of Saudi Arabia (Approval# 8/25/454266, 30 September 2013).

### 2.2. Biochemical Analysis

Fasting blood samples were collected. Circulating levels of RANKL and OPG were determined by multiplex assay kits using the Luminex^®^ xMAP^®^ Technology platform (Luminex Corporation, Austin, TX, USA) (cat# HBNMAG-51K). Serum NTx was determined using ELISA assay (Alere Scarborough, Inc.; 10 Southgate Road, Scarborough, ME, USA). Osteocalcin was measured using electrochemiluminescence immunoassay using Roche Cobas-e411 kit (Roche Diagnostics, Mannheim, Germany). COBAS e411 analyzer also measured testosterone and estradiol by a standard protocol provided along with the kit (Roche Diagnostics).

### 2.3. Data Analysis

Data were analyzed using the Statistical Package for Social Sciences (SPSS) 22.0 (SPSS, Inc., Chicago, IL, USA). Continuous data were presented as mean ± standard deviation (SD) for normal variables, and non-Gaussian variables were presented in median (1st and 3rd) percentiles. All categorical variables were presented as frequency and percentages. One-way ANOVA and Kruskal–Wallis H test were performed to check the mean and median differences. Pearson’s and Spearman’s correlations were performed for normal and non-normal variables, respectively, to find the bivariate associations between the variables. Stepwise linear regression analysis was performed for T-score and BMD as dependent variables; and RANKL, OPG, and RANKL/OPG were independent predictors. *p*-value < 0.05 was considered statistically significant.

## 3. Results

### 3.1. Clinical Characteristics of Study Groups

The general characteristics of the participants according to the study groups were presented as Table 1. The osteoporosis group was significantly older than the normal group (*p* = 0.003). Years since menopause was also significantly higher in osteoporosis group compared to osteopenia group and both were significantly higher than the normal group (*p* < 0.001). In addition, BMI was significantly lower in the osteoporosis group than in the normal group (*p* < 0.001). As expected, spine BMD, femoral neck BMD, and T-score (spine) was significantly lower in the osteoporosis group than in normal and osteopenia groups (*p* < 0.001). No significant differences were observed in the levels of osteocalcin, NTx, RANKL, OPG and the RANKL/OPG ratio, estradiol, and testosterone.

Figure 1 shows the circulating levels of RANKL and OPG in the three study groups in our data.

### 3.2. Bivariate Correlations of RANKL, OPG, and the RANKL/OPG Ratio with Other Parameters

Table 2 RANKL levels showed no significant correlation with age, T-score (spine), BMD spine, femoral neck right BMD, and femoral neck right BMD in any groups. In contrast, RANKL was inversely and significantly correlated with NTx in the osteopenia group (*r* = −0.37, *p* < 0.05). OPG, on the other hand, had a significant positive correlation with age in the osteoporosis group (*r* = 0.29, *p* < 0.05), while it was inversely correlated with femoral neck left BMD (*r* = −0.56, *p* < 0.001) and femoral neck right BMD (*r* = −0.37, *p* < 0.05) in the same group. OPG levels showed no significant correlation with T-score (spine) and BMD spine in any study groups. In addition, no significant correlation was observed between OPG and NTx. The RANKL/OPG ratio showed positive and significant correlation with BMI (*r* = 0.34, *p* < 0.05), T-score dual left (*r* = 0.37, *p* < 0.05), femoral neck left BMD (*r* = 0.36, *p* < 0.05), and femoral neck right BMD (*r* = 0.35, *p* < 0.05) in the osteopenia group. While it was inversely and significantly correlated with WHR in the osteoporosis group (*r* = −0.38, *p* < 0.05). Sex hormones, testosterone, and estradiol, revealed a strong positive and significant correlation with OPG in the normal group (*r* = 0.58 and *r* = 0.55, respectively, *p* < 0.01), but no significance was found in the other two groups. To further explore the association of age with OPG and RANKL, subjects were stratified into tertiles according to age and found no differences in subgroups with respect to OPG and RANKL (Appendix A).

The bivariate correlations of RANK and OPG levels with BMD (femoral neck) were plotted as Figure 2.

### 3.3. Stepwise Regression Analysis

Multiple regression analysis was performed to assess the influence of RANKL, OPG, and the RANKL/OPG ratio on measured T-scores and BMDs, and the results were presented in Table 3. The results showed that OPG was a contributing factor for femoral neck BMD (*p* = 0.009) in our study of postmenopausal women overall. On the other hand, RANKL has no significant effect on T-scores or BMDs in the three groups.

## 4. Discussion

OPG and RANKL are important markers in bone metabolism. However, limited studies have been published where the relationship between the circulating levels of these markers and BMD has been investigated, especially in Arab postmenopausal women. This study thus looked at the relationships between BMD and serum levels of RANKL, OPG, and the RANKL/OPG ratio in 617 postmenopausal Saudi women. In addition to the interesting associations of these circulating markers with some anthropometric indices, this study suggested a significant inverse relationship between OPG and BMD, especially at the femoral neck in postmenopausal osteoporotic Arab women. Besides, a significant inverse association of RANKL with NTx suggested the role of RANKL in bone remodeling in postmenopausal women. Associations of the menopausal age with these markers of bone metabolism also suggest a modulatory effect of years since menopause and female sex hormones in bone metabolism.

Our findings showed that serum OPG levels increased significantly with age in this study, which is consistent with earlier findings in various populations [30,31,32]. Other studies, however, found that serum levels of OPG were age-independent and did not show differences between pre-and postmenopausal women [33]. These discrepancies might be due to differences in sample size, the elevated bone turnover, and estrogen deficiency in postmenopausal women. Our findings revealed no significant differences in serum RANKL concentrations or RANKL/OPG ratios across all groups, implying that OPG and RANKL production are regulated by different mechanisms [25].

Many studies have investigated serum OPG effects on BMD and bone turnover markers with varying outcomes [21,22,23]. In some studies, BMD was positive, either inversely associated or not correlated with OPG serum levels in postmenopausal women [21,34]. An inverse significant correlation between OPG and femoral neck BMD in the osteoporosis group was found in our study. Data from previous studies showed inconsistency in the effects of RANKL and OPG on BMD and bone turnover markers. A study by Mezquita et al. found that OPG and RANKL were independently correlated to vertebral fractures and osteoporosis [35]. Stern et al. observed that OPG had a significant positive correlation with BMD, and found RANKL to be inversely associated with BMD only in men [36]. Nabipour et al. found that serum levels of OPG and the RANKL/OPG ratio were significantly correlated with age; moreover, age, RANKL, and OPG were independent determinants of BMD [37].

Our results showed that RANKL and the RANKL/OPG ratio were inversely correlated with NTx. Other studies have revealed similar correlations between OPG or RANKL and bone turnover markers. For example, some have shown inverse associations between OPG and osteocalcin [34] and found it a good predictor of osteoporosis and bone turnover markers [38]. The association of RANKL with NTx in this study may seem to be confusing since, between the three study groups based on BMD (normal, osteopenia and osteoporosis), the circulating levels of RANKL showed no significant difference. However, a *p*-value of 0.07 is close to statistical significance meaning that there are differences between the groups even though we do not report this in this study. This may explain the discrepancies found in the literature on the association of RANKL and OPG with BMD. Some studies demonstrated OPG favors bone formation [39,40]. In contrast, some studies found no correlations between BMD and serum levels of bone turnover biochemical markers with RANKL or OPG [41,42]. Furthermore, Chiba et al. and Liu et al. did not find a significant association for serum RANKL levels with BMD in postmenopausal women [38,40]. These discrepancies in results from the previous studies could be partially explained by the variations in the available ELISA assays. Furthermore, increases in OPG will be expected to increase bone loss as part of the mechanism to minimize bone turnover and thus decrease bone loss. Consequently, it is likely for the associations between bone turnover, BMD, RANKL, and OPG to be much more complex. For instance, several studies found increasing OPG with age, suggesting a compensatory response to increased osteoclastic resorption [37,40].

Our results showed a positive correlation between sex hormones and OPG in normal BMD participants. The in vitro stimulation of OPG biosynthesis in osteoblasts cell lineage by estrogen gives great evidence for the contribution of the RANKL/RANK/OPG system in the sex steroid hormones bone-sparing effects [43,44,45,46] as well as in human subjects [46]. However, OPG stimulation mediated by estrogen might inhibit bone resorption, a mechanism not utilized by androgens [46,47]. Furthermore, RANKL trafficking to the pre-osteoblast membrane might be inhibited by sex steroid hormones, accelerating its shedding or internalization. Estrogens also stimulate OPG expression [44,46,48] other than regulating RANKL membrane association in pre-osteoblasts [49].

In this study, multiple regression analyses showed no significant effect for the measured bone markers on BMD or t-scores except the inverse association of the levels of OPG with femoral neck BMD. Although, RANKL and OPG may not be specific to BMD, some probabilities might explain this. First, in addition to bone, a diversity of other tissues can express OPG mRNAs such as heart, lung, and kidney. OPG has been associated with vascular calcification [50] and diabetic microvascular complications [51]. As such, RANKL and OPG may serve as a compensatory protecting mechanism in contrast to age-related disorders, even if the alteration in these biochemical markers may not be specific to bone metabolism [22]. Second, vitamin D status which is associated with these bone markers and highly prevalent in the cohort as early as reproductive age, was not assessed [52]. In early postmenopausal women, RANKL surface levels per bone marrow mononuclear cells were 2–3 fold higher than premenopausal and estrogen-treated postmenopausal women and displayed no association with its serum levels [31].

The strengths of the current study include, firstly, a sizable study sample of 617 participants. It gives this study an advantage over several other studies with fewer participants. Secondly, this is a multi-center recruitment study, which appropriately represents a population. Thirdly, to our knowledge, this is the first study to report the associations of OPG and RANKL with BMD in a cohort of Arab postmenopausal women. However, the authors acknowledge certain limitations. The study is limited because it was restricted to postmenopausal women only. Therefore, the results cannot be extrapolated to other populations. It is also a cross-sectional study, limiting its applicability outside the proposed relationship’s cause and effect. A longitudinal study in this population would be interesting and needed to investigate such relationship. Furthermore, calcium intake and vitamin D status of the participants was not taken into consideration and may have impacted the results.

## 5. Conclusions

In summary, serum levels of RANKL were not associated with BMD among Arab postmenopausal women and OPG had a significant inverse association with BMD, especially at the femoral neck in osteoporotic Arab women. RANKL showed significant inverse association with NTx, a marker of bone resorption. The results indicate that menopausal age and sex hormones may affect these bone markers suggesting a role in bone metabolism. The relative increase in serum OPG concentration with age and menopause is likely to be a protective reaction for elevated bone resorption. This needs to be investigated prospectively.

## Figures and Tables

**Figure 1 medicina-58-00976-f001:**
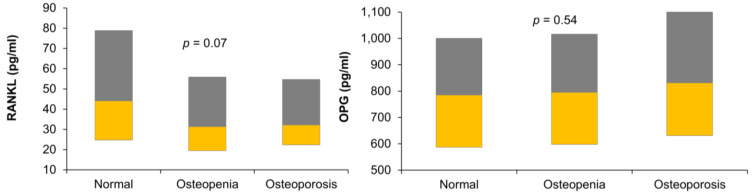
Boxplot showing the levels of receptor activator of nuclear factor-kappa B ligand (RANKL) and osteoportegerin (OPG) in the three study groups.

**Figure 2 medicina-58-00976-f002:**
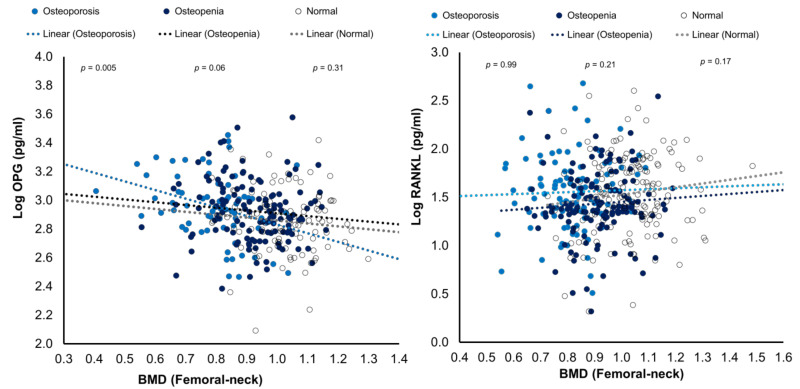
Scatterplots showing the associations of RANKL and OPG with BMD (femoral neck) in the three study groups.

**Table 1 medicina-58-00976-t001:** Clinical characteristics of participants.

Parameters	Normal	Osteopenia	Osteoporosis	*p*-Value
*n*	169	282	169	
Age (year)	56.1 ± 6.1	57.7 ± 6.9	58.7 ± 7.5 ^A^	0.003
Age of menarche	13.1 ± 1.4	13.4 ± 1.5	13.5 ± 1.7	0.08
Age at first pregnancy	19.3 ± 3.7	18.6 ± 3.3	19.4 ± 4.0	0.04
Years since menopause	6.2 ± 4.9	8.5 ± 6.0 ^A^	10.1 ± 7.9 ^AB^	<0.001
BMI (kg/m^2^)	34.0 ± 5.7	33.4 ± 5.5	30.3 ± 6.2 ^A^	<0.001
WHR	0.92 ± 0.11	0.92 ± 0.09	0.92 ± 0.09	0.85
BMD (spine)	1.16 ± 0.08	0.97 ± 0.9 ^A^	0.83 ± 0.08 ^AB^	<0.001
BMD (femoral neck left)	1.04 ± 0.09	0.90 ± 0.8 ^A^	0.80 ± 0.13 ^AB^	<0.001
T-score (spine)	−0.03 ± 0.7	−1.68 ± 0.5 ^A^	−3.04 ± 0.6 ^AB^	<0.001
RANKL (pg/mL)	44 (24.8–78.9)	31.2 (19.4–55.9)	32.1 (22.4–54.7)	0.07
OPG (pg/mL)	784.1 (587–1001)	794.4 (597.8–1016)	830.5 (631.3–1102)	0.54
RANKL/OPG	0.04 (0.03–0.09)	0.04 (0.02–0.06)	0.04 (0.02–0.05)	0.48
Osteocalcin (ng/mL)	11.7 (3.6–16.7)	8.5 (2.8–13.7)	10.0 (3.8–13.0)	0.10
NTx (nmol)	57.8 (45–73.4)	54.3 (41.4–69.2)	58.4 (40.9–77.2)	0.91
Testosterone (ng/mL)	0.8 (0.5–1.4)	0.6 (0.3–1.0)	0.7 (0.3–1.0)	0.09
Estradiol (pg/mL)	69.3 (39.7–183.1)	63.7 (34.2–195.7)	59.6 (34.2–205.3)	0.93

Note: Data presented as mean ± SD and median (25th–75th) percentiles for Gaussian and non-Gaussian variables. The superscripts A and B mean significant difference compared to normal and osteopenia groups, respectively. BMI, body mass index; BMD, bone mineral density; OPG, osteoprotegerin; RANKL, receptor activator of nuclear factor-kappa B ligand; WHR, waist-hip ratio. A *p*-value was considered significant at 0.05 and 0.01 levels.

**Table 2 medicina-58-00976-t002:** Significant correlations of RANKL, OPG, and RANKL/OPG ratio with other parameters measured.

Parameters	All			Normal		Osteopenia			Osteoporosis		
	RANKL	OPG	RANKL/OPG	RANKL	OPG	RANKL	OPG	RANKL/OPG	RANKL	OPG	RANKL/OPG
Age (year)		0.20 **								0.29 *	
Age at first pregnancy									−0.34 **		
BMI (kg/m^2^)								0.34 *			
WHR											−0.38 *
T-score AP spine											
femoral neck left BMD		−0.27 *	0.37 **					0.36 *		−0.56 **	
NTx (nmol)						−0.37 *		−0.41 *			
Testosterone (ng/mL)		0.19 *			0.58 **						
Estradiol (pg/mL)		0.21 *			0.55 **						

Note: Only significant correlations were presented. Data presented as coefficient (R); * denotes significance at 0.05 level; ** denotes significance at 0.01 level.

**Table 3 medicina-58-00976-t003:** Stepwise regression analysis: T-score and BMD as dependent parameters.

Parameters	All Participants	
	β (SE)	*p*-Value
T-score (spine)		
RANKL	0.05 (0.55)	0.93
OPG	−0.96 (0.81)	0.23
RANKL/OPG	−0.58 (2.30)	0.81
BMD (spine)		
RANKL	0.12 (0.10)	0.38
OPG	−0.14 (0.2)	0.39
RANKL/OPG	−0.60 (0.4)	0.1
BMD (femoral neck)		
RANKL	0.16 (0.10)	0.09
OPG	−0.32 (0.12)	0.009
RANKL/OPG	−0.25 (0.50)	0.6

Note: Data presented as beta-coefficient (β) and standard error (SE). *p*-value < 0.05 is considered significant.

## Data Availability

The data presented in this study are available on request from the corresponding author.

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
