# Peer review of "Associations of Bone Mineral Density with RANKL and Osteoprotegerin in Arab Postmenopausal Women: A Cross-Sectional Study"

_medicina, 2022, doi:10.3390/medicina58080976_

Round 1

Reviewer 1 Report

This study investigated the relationship between circulating levels of OPG and RANKL with BMD in Arab postmenopausal women.

1)The study did not observe the difference between OPG and RANKL among groups as shown in Figure 1. On the basis of no difference among each group, analyzing the correlation will greatly reduce the significance of the study. 

2)The production of charts is not standardized. There is no figure legends for figure 1 and 2. Table 2 should list all the relevant coefficients.

3) How to explain that RANKL is related to NTX but not BMD?The authors should disscuss these issues.

4) pg/ml should be pg/mL

Author Response

This study investigated the relationship between circulating levels of OPG and RANKL with BMD in Arab postmenopausal women.

1)The study did not observe the difference between OPG and RANKL among groups as shown in Figure 1. On the basis of no difference among each group, analyzing the correlation will greatly reduce the significance of the study. 

Response: The authors thank the reviewer for the suggestion and agree that there was no statistical significance observed in circulating levels of either RANKL or OPG between the groups (normal, osteopenia and osteoporosis). The authors still explored within group associations with BMD and found that OPG is negatively associated with Femoral-Neck BMD which was confirmed with a step-wise regression analysis. With this finding, the authors concluded that the increase in OPG levels with age and menopause might be a protective mechanism for elevated bone resorption and consequently low BMD in post-menopausal women. However, this needs to be further verified with longitudinal studies. Since this was an interesting finding in this study, the authors feel to retain the within-group correlation analysis in spite of no statistical difference between groups. The lack of correlation still has merit as it denotes differences in the associations of bone markers with bone density relative to ethnicity.   

2)The production of charts is not standardized. There is no figure legends for figure 1 and 2. Table 2 should list all the relevant coefficients.

Response: Legends have now been provided in the figures. In table 2, only significant correlations were listed, the rest of the associations were not written. This is to facilitate readers in interpreting the table, as writing all the correlations, significant or not, would appear busy.

3) How to explain that RANKL is related to NTX but not BMD? The authors should discuss these issues.

Response: The authors agree with the reviewer that this finding may seem to be confusing since between the three study groups based on BMD (normal, osteopenia and osteoporosis), the circulating levels of RANKL showed no significant difference. However, a p-value of 0.07 is close to statistical significance meaning that there are differences between the groups even though we do not report in this study. This may explain the discrepancies found in the literature on the association of RANKL and OPG with BMD. This has been explained and added in the revised discussion section.

4) pg/ml should be pg/mL

Response: Thanks, this has been changed in the revised draft

Reviewer 2 Report

The Authors investigated the associations between circulating levels of osteoprotegerin (OPG) and receptor activator of nuclear factor kappa-Β ligand (RANKL) with bone mineral density (BMD) in Arab postmenopausal women. RANKL/OPG ratio was positvely associatad with femoral BMD in the osteopenia patients, and at multiple regression analysis OPG was independently associated to BMD variations.

This is an interesting paper focused on a rilevant topic. Some points need to be furthers discussed.

Few clinical details about participants are shown. Particularly, diseases potentially impacting on bone health.

The Authors say to have excluded women under medications that can affect BMD and/or bone turnover markers for at least 6 months. Further details are needed (e.g. the effects of a single zoledronic acid dose on bone and surrogate markers of bone turn-over may be observed over 6 month). Which drugs they considered as exclusion criteria? Even pulsed electromagnetic fields (PEMFs) have been proven to modulate RANKL/OPG system in postmenopausal women (see “Bone 2018 Nov;116:42-46. doi: 10.1016/j.bone.2018.07.010.”), thus its use should be mentioned as exclusion criteria.

Women under vitamin D or calcium supplementation were excluded. Which was the vitamin D status and calcium intake of these women? And of participants? Calcium intake and vitamin D status if poor may exert a negative feedback on PTH levels with consequences on RANK-L/OPG.   

Did the Authors investigated history of fractures?

Did the Authors consider stratification of participants by age?

Author Response

The Authors investigated the associations between circulating levels of osteoprotegerin (OPG) and receptor activator of nuclear factor kappa-Β ligand (RANKL) with bone mineral density (BMD) in Arab postmenopausal women. RANKL/OPG ratio was positively associated with femoral BMD in the osteopenia patients, and at multiple regression analysis OPG was independently associated to BMD variations.

This is an interesting paper focused on a relevant topic. Some points need to be furthers discussed.

Response: The authors thank the reviewer for the appreciation and acknowledge that this revision from both the reviewers have greatly improved the quality of the manuscript.

Few clinical details about participants are shown. Particularly, diseases potentially impacting on bone health.

The Authors say to have excluded women under medications that can affect BMD and/or bone turnover markers for at least 6 months. Further details are needed (e.g. the effects of a single zoledronic acid dose on bone and surrogate markers of bone turn-over may be observed over 6 month). Which drugs they considered as exclusion criteria? Even pulsed electromagnetic fields (PEMFs) have been proven to modulate RANKL/OPG system in postmenopausal women (see “Bone 2018 Nov;116:42-46. doi: 10.1016/j.bone.2018.07.010.”), thus its use should be mentioned as exclusion criteria.

Response: The authors agree with the reviewer that the categories listed should be mentioned in the exclusion criteria. This has been done in the revised draft, thanks.

Women under vitamin D or calcium supplementation were excluded. Which was the vitamin D status and calcium intake of these women? And of participants? Calcium intake and vitamin D status if poor may exert a negative feedback on PTH levels with consequences on RANK-L/OPG. 

Response: The authors agree with the reviewer that low calcium intake and poor vitamin D status may exert a negative feedback on PTH level. However, unfortunately calcium intake and vitamin D status of the participants was not taken into consideration and may have impact on the results. This has been acknowledged as a limitation in the revised draft.  

Did the Authors investigated history of fractures?

Response: The questionnaire filled up by the participants at the time of recruitment had questions like whether they had fructures in the last six months and those who had were excluded from the study.

Did the Authors consider stratification of participants by age?

Response: Within the studied groups, the correlation analysis of age with RANKL, OPG and RANKL/OPG revealed only a significant positive association of OPG with age. This has been mentioned and discussed in the manuscript. We have performed additional analysis stratifying age according to tertiles and found no significant differences with parameters of interest. This has now been included as supplementary tables.   

Round 2

Reviewer 1 Report

At present, the article has been satisfactory.